# Subgrouping a Large U.S. Sample of Patients with Fibromyalgia Using the Fibromyalgia Impact Questionnaire-Revised

**DOI:** 10.3390/ijerph18010247

**Published:** 2020-12-31

**Authors:** Adrián Pérez-Aranda, Albert Feliu-Soler, Scott D. Mist, Kim D. Jones, Yolanda López-Del-Hoyo, Rebeca Oliván-Arévalo, Anna Kratz, David A. Williams, Juan V. Luciano

**Affiliations:** 1Department of Basic, Evolutive and Educational Psychology, Faculty of Psychology, Universitat Autònoma de Barcelona, 08193 Cerdanyola del Vallés, Spain; 2AGORA Research Group, Teaching, Research & Innovation Unit, Parc Sanitari Sant Joan de Déu, 08830 St. Boi de Llobregat, Spain; rebeca.alberuela@gmail.com; 3Institute of Neuropsychiatry and Addictions, Hospital del Mar, 08003 Barcelona, Spain; 4Centro de Investigación en Red de Salud Mental, CIBERSAM, 28029 Madrid, Spain; 5Institut de Recerca Sant Joan de Déu, 08950 Esplugues de Llobregat, Spain; 6Fibromyalgia Research and Treatment Group, Department of Anesthesiology & Perioperative Medicine, Oregon Health & Science University, Portland, OR 97239, USA; mists@ohsu.edu; 7School of Nursing, Linfield University and School of Medicine, Oregon Health & Science University, Portland, OR 97239, USA; kim.jones@linfield.edu; 8Department of Psychology and Sociology, Univesity of Zaragoza, 50001 Zaragoza, Spain; ylopez.iacs@gmail.com; 9Instituto de Investigación Sanitaria Aragón, 50009 Zaragoza, Spain; 10Department of Physical Medicine and Rehabilitation, University of Michigan, Ann Arbor, MI 48109, USA; alkratz@med.umich.edu; 11Departments of Anesthesiology, Internal Medicine, Psychiatry, and Psychology, University of Michigan, Ann Arbor, MI 48109, USA; daveawms@med.umich.edu

**Keywords:** Fibromyalgia Impact Questionnaire Revised, Fibromyalgia, clusters, latent profile analysis, hierarchical cluster analysis

## Abstract

Fibromyalgia (FM) is a heterogeneous and complex syndrome; different studies have tried to describe subgroups of FM patients, and a 4-cluster classification based on the Fibromyalgia Impact Questionnaire-Revised (FIQR) has been recently validated. This study aims to cross-validate this classification in a large US sample of FM patients. A pooled sample of 6280 patients was used. First, we computed a hierarchical cluster analysis (HCA) using FIQR scores at item level. Then, a latent profile analysis (LPA) served to confirm the accuracy of the taxonomy. Additionally, a cluster calculator was developed to estimate the predicted subgroup using an ordinal regression analysis. Self-reported clinical measures were used to examine the external validity of the subgroups in part of the sample. The HCA yielded a 4-subgroup distribution, which was confirmed by the LPA. Each cluster represented a different level of severity: “Mild–moderate”, “moderate”, “moderate–severe”, and “severe”. Significant differences between clusters were observed in most of the clinical measures (e.g., fatigue, sleep problems, anxiety). Interestingly, lower levels of education were associated with higher FM severity. This study corroborates a 4-cluster distribution based on FIQR scores to classify US adults with FM. The classification may have relevant clinical implications for diagnosis and treatment response.

## 1. Introduction

Fibromyalgia (FM) is a chronic pain condition that affects approximately 2% of the general population worldwide [1]. Patients present a wide range of symptoms, primarily chronic widespread pain, allodynia and hyperalgesia, stiffness, fatigue, sleep problems, impaired balance, cognitive difficulties, and distress [2,3]. Moreover, FM often co-occurs with psychiatric disorders and other conditions characterized by central sensitization [4,5].

Currently, FM management focuses on symptom reduction through prescription of different medications and recommendation of psychotherapies and other non-pharmacological interventions such as aerobic exercise [6]. However, none of these provide more than a modest impact on symptom relief and quality of life [7,8,9,10]. Heterogeneity in treatment response may obscure positive findings for some people with FM; thus, describing subtypes could be a valuable contribution for moving toward the ability to match treatments to individuals based on their personal characteristics and clinical status.

There has been a great deal of work attempting to identify FM subgroups [11]. However, not all studies have used the same methodology; while some have used self-report measures, others have included variables such as the body mass index or biological measures to classify FM [12]. There are many approaches to identifying FM subgroupings, each holding potential heuristic and clinical value.

De-Souza et al. [13] used the Fibromyalgia Impact Questionnaire (FIQ) [14] as a cluster-derivation measure. This approach to FM classification is clinically relevant and can be easily implemented in real world clinical practice. De-Souza et al. observed two subgroups, mainly differentiated by the impact of morning tiredness, anxiety, and depressive symptoms. Subsequent studies used this approach in FM samples from different countries [15,16,17], observing additional between-cluster differences in number of comorbidities, health-related quality of life, and sleep quality.

More recently, the revised version of the questionnaire (i.e., FIQR) [18] has been used to classify subtypes of patients in Italy [19] and Spain [11]. The FIQR represents a refinement of the original FIQ; the scoring system is easier, and the third domain (i.e., Symptom severity) includes new items. Perhaps due to these modifications, the clustering studies conducted in Italy and Spain based on the FIQR have found more than two clusters. Salaffi et al. [19] found three subgroups in a sample of 353 Italian FM patients: “Mild”, “moderate”, and “severe”. However, no other measures were included in the study to assess the external validity of this classification. Pérez-Aranda et al. [11] analyzed a sample of Spanish FM patients (N = 947) and found 4 clusters: “mild-moderate”, “moderate”, “moderate-severe”, and “severe”. These clusters presented significant differences in clinical variables (e.g., anxiety, depressive symptoms, fibrofog, and stress) and in indirect costs, reflecting the socioeconomic impact of severity of FM. This four-cluster classification has not been replicated in other cultures, so its global applicability remains unclear.

The present study provides a cross-validation of the four-cluster classification based on FIQR scores in a large US sample (N = 6280) of individuals with FM. In addition, we developed a publicly accessible online cluster calculator to make the present results easily implementable in US clinical settings.

## 2. Materials and Methods

### 2.1. Participants

For the purpose of this study, we pooled two large US samples of individuals with FM. One subsample consists of 1639 adult FM patients who participated in a study to evaluate Patient Reported Outcomes Measurement Information System (PROMIS) measures in FM [20]. All participants were members of the National Fibromyalgia Association (NFA), a patient advocacy organization. Most of the participants were female (96.9%), middle-aged (M = 52.8, SD = 10.57, Range = 20–82), and Caucasian (79.9%). The study protocol was approved by the University of Michigan Medical Institutional Review Board; all data were collected via online surveys between April 2009 and May 2010, and responses were deidentified. For the present study, 56 cases were excluded due to missing FIQR data. This study included clinical self-report measures that were used for the assessment of external validity of the subgroups in the present study (see measures used for external validation); random subsamples of around 250–280 participants were generated to answer each self-reported measure.

The other subsample consisted of 4986 adult FM patients who took part in a study designed to examine the relationship between mindfulness facets and FM impact [21]. In this study, the FM diagnosis was self-reported by participants, and the responses to the questionnaires were given via an online survey, after Oregon Health & Science University Institutional Review Board approval. No missing data were found in our main study measure (FIQR). The survey asked about gender, age, marital status, and education level, among other sociodemographic information. Participants represented all 50 states in the US, with some participants residing in other countries (n = 289); these latter cases were excluded in the present study. Most participants were women (96.6%) with a mean age of 52.2 (SD = 10.6, Range = 21–89), and their FM symptoms had been present for more than 10 years in most cases (74%). Almost half of the participants reported university studies (47%), and 53% of the sample were not working when they were evaluated.

### 2.2. Study Measures

#### 2.2.1. Measure Used for Cluster Derivation: The FIQR

The FIQR is the gold standard for evaluating functional status of patients with FM. It consists of 21 items measuring the core symptoms and impact of FM in the last 7 days and contains three subscales: Physical dysfunction (0–30), overall impact (0–20), and intensity of the symptoms (0–50). Every item is rated using a Likert scale (0 = “lowest severity” to 10 = “highest severity”). The global score is obtained by adding the three subscales scores (0–100). The FIQR has high internal consistency (α = 0.91–0.95), good test-retest reliability (r = 0.82), and construct validity [22]. The FIQR was completed by a total of 6280 respondents.

#### 2.2.2. Measures Used for Clinical External Validation of the FIQR Profiles

The Brief Pain Inventory-short form (BPI-SF) [23] is a 9-item measure that assesses the intensity and impact of pain. Pain intensity is assessed at its “worst”, “least”, “average”, and “now”; for the present study, a composite of the four pain items was used as the mean severity score (ranging from 0 = “no pain” to 10 = “pain as bad as one can imagine”). Pain interference is scored as the mean of the 7 interference items and ranges from 0 (pain does not interfere) to 10 (it completely interferes). The inventory presents strong psychometric properties [24]. It was completed by 255 participants of the first subsample (4.1% of the pooled sample).

The Hospital Anxiety and Depression Scale (HADS) [25] is a 14-item self-reported scale developed to assess the presence of anxiety and depression symptoms during the last 7 days. The total score ranges from 0 to 42, and each subscale (HADS-Anxiety and HADS-Depression) includes 7 items whose scores range from 0 (lowest severity) to 21 (highest severity). This scale has shown sound psychometric properties [26]. The HADS was completed by 285 participants of the first subsample (4.5% of the pooled sample).

The Multiple Ability Self-Report Questionnaire (MASQ) [27] is a 38-item questionnaire that has been used to characterize perceived cognitive impairment (i.e., “fibrofog”) and is commonly used to assess cognitive dysfunction in FM clinical trials [20]. The questionnaire explores everyday performance in 5 domains: Language, visual-perceptual abilities, verbal and visual memory, and attention/concentration. Each subscale contains 8 items, except the visual-perceptual ability subscale, which contains 6. The items are rated on a 5-point Likert scale (1 = almost never; 5 = almost always) regarding the frequency of specific difficulties. The total score ranges from 38 to 190, where higher scores indicate more severe perceived impairment [28]. The MASQ was completed by 252 participants of the first subsample (4% of the pooled sample).

The Multidimensional Fatigue Inventory (MFI-20) [29] is a 20-item measure that consists of 5 subscales: General fatigue, physical fatigue, activity, motivation, and mental fatigue. Each item is scored in a 5-point Likert scale, and the score of each subscale ranges from 4 to 20, with higher scores reflecting greater severity. The reliability and validity of this inventory is adequate. The MFI-20 was completed by 274 participants of the first subsample (4.4% of the pooled sample).

The Medical Outcomes Study sleep scale (MOS-Sleep) [30] is a 12-item measure which can be divided into 6 dimensions; however, for the present study, only the sleep problem index (i.e., a summary score of 6 items) was used. Each of these 6 items is scored on a Likert-scale (1 = all the time; 6 = none of the time). The score of the sleep problem index ranges from 6 to 36, where higher scores indicate worse sleep functioning. The MOS-Sleep presents strong psychometric properties [31]. This measure was completed by 266 participants of the first subsample (4.2% of the pooled sample).

### 2.3. Statistical Analyses

Clusters were established by means of exploratory hierarchical agglomerative cluster analysis (HCA), using the Ward method to form clusters, which is the most widely used method in similar studies. The software used for performing these analyses was SPSS v22.0 (SPSS Inc., Chicago, IL, USA). Following Pérez-Aranda et al.’s methodology [11], a latent profile analysis (LPA) was computed to confirm the optimal number of subgroups. LPA is a person-centered statistical technique that takes measurement error into account and, thus, represents a potentially more statistically sophisticated approach [32]. We used MPlus v7.4 (Mñuthen & Múthen, Los Angeles, CA, USA) to compute LPA.

An ordinal regression analysis was carried out to determine the predicted cluster membership [33]. For this analysis, FIQR items were used as independent variables and the cluster category was taken as the dependent variable. The ordinal regression would then estimate the probability of each cluster for every case, selecting the one with the highest probability. Then, the Cohen’s kappa (k) coefficient was calculated to measure the agreement between the predicted assigned subgroup and the originally assigned (k rule of thumb: ≤ 0 = no agreement, 0.01–0.20 = none to slight, 0.21–0.40 = fair, 0.41–0.60 = moderate, 0.61–0.80 = substantial, and 0.81–1.00 = almost perfect agreement) [34].

Between-group differences in sociodemographic data and the clinical measures were analyzed in order to assess the external validity of the calculator-derived classification. One-way ANOVAs or MANOVAs with Games-Howell post hoc tests were computed for continuous variables (i.e., FIQR items and the scores of the rest of scales), and the chi squared test was used for categorical data. The effect size in analyses of variance was based on partial eta-squared (ηp^2^ rule of the thumb: 0.01 = small; 0.06 = medium; and 0.14 = large) (10), which can be interpreted as the proportion of variance in the dependent variable that is attributable to each effect.

## 3. Results

### 3.1. Hierarchical Cluster Analysis

The HCA distributed the sample in 4 subgroups of patients with different levels of FM severity, which accounted for 17.6%, 24.2%, 23.4%, and 34.7% of the sample, respectively. The pairwise comparisons showed statistically significant differences between clusters for all the FIQR items (all *p* values < 0.001), always following the same pattern: 1 < 2 < 3 < 4. Thus, the subgroups could be labelled as in a previous study [11]: “Mild–moderate”, “moderate”, “moderate–severe”, and “severe” impact of FM. Figure 1 shows the distribution of the FIQR items’ mean score for each calculator-derived cluster.

The level of agreement between the clusters obtained in the HCA and the ones predicted by the calculator was substantial (k = 0.77). In this case, the predicted clusters accounted for 11.6%, 35.9%, 23.9%, and 28.6% of the sample. Therefore, we decided to retain the calculator-derived clusters for further analyses (external validity).

### 3.2. Latent Profile Analysis

The LPA indicated that the four-profile solution fit our data better than the other profile solutions (see Appendix A). Entropy values indicated that patients were well classified in all class solutions overall. The LMR-LRT *p* values suggested that the 5-class model was not superior to the 4-class model. Good discrimination between classes was observed with high-average latent class probabilities for most likely latent class membership (class 1 = 0.96; class 2 = 0.93; class 3 = 0.94; and class 4 = 0.95). The four subgroups obtained with the LPA represented 11.8%, 23.5%, 36.2%, and 28.5% of the sample, respectively. The pairwise comparisons showed statistically significant differences between clusters for all the FIQR items (all *p* values < 0.001), always following the same pattern: 1 < 2 < 3 < 4. The comparison between the two methods (HCA and LPA) showed a substantial level of agreement (k = 0.73, *p* < 0.001). Subgroup 3 was the one with the highest level of agreement (96.3%), followed by subgroup 4 (80.5%), subgroup 2 (73.2%), and subgroup 1 (68.1%). Appendix A shows the distribution of the FIQR items’ mean score for each profile.

### 3.3. Between-Cluster Differences in Sociodemographic and Clinical Variables

Table 1 presents the sociodemographic characteristics of the subgroups using the calculator predicted cluster per each case. Differences between subgroups were observed in age (*p* < 0.001), gender (χ² = 17.01, *p* = 0.009), marital status (χ² = 29.41, *p* = 0.001), and years since FM diagnosis (F = 4.01, *p* = 0.007), although the effect sizes were small (ηp^2^ < 0.006). In contrast, there was a large effect for differences in distribution across clusters for education level, which was only assessed in the second sample (χ² = 261.08, *p* < 0.001).

Table 2 presents the results for all the self-reported measures using the predicted cluster per each case. All of them presented statistically significant differences between clusters with large effect sizes in all cases but three: For HADS-A and MASQ, the effect size was medium (i.e., ηp^2^ between 0.06 and 0.14), and the HADS-D showed no differences between clusters (*p* = 0.989). The pairwise comparisons showed the expected relation between clusters (i.e., 1 < 2 < 3 < 4) in all the FIQR subscales and total score as well as in the BPI. In the remaining outcomes, there can be observed a tendency towards this relation, yet some of the pairwise comparisons did not present significant differences (i.e., MASQ, HADS-A, MFI-20, and MOS-Sleep).

## 4. Discussion

The 4-cluster solution based on FIQR scores at item level obtained in a large Spanish sample of patients with FM [11] seems promising due to its easily applicability in clinical practice, but it lacked cross-cultural validity. The present study fills this gap by replicating the proposed classification in a large US sample of individuals with FM.

Due to its consideration as “gold standard” measure of functional status, the FIQR has been increasingly used in FM research, not only as a clinical outcome, but also as a possible measure of classification of FM subtypes. The main advantage of using this classification system is that it only requires one self-report measure that is not time consuming (FIQR can be completed by patients in < 2 min), unlike previous studies that have used a comprehensive battery of instruments and different biomarkers [12].

Nevertheless, this classification presents some limitations that need to be acknowledged: (1) It only provides a general description of the functional impact of FM, but not detailed information on other relevant variables (e.g., history of childhood maltreatment, comorbidities, biological markers, personality traits). This lack of detail may be balanced however, by the clinical efficiency offered by this relatively brief classification method. (2) The studies that have used this method to classify FM patients have reported different cluster solutions: While the Italian sample (N = 353) analyzed by Salaffi et al. [19] was classified into 3 severity clusters, the Spanish sample (N = 946) analyzed by Pérez-Aranda et al. [11] yielded 4 clusters. Considering the cultural and sociodemographic similarities between the two samples, it is possible that the different results are due to the size of the samples, i.e., hypothetically, a larger Italian sample would have been equally distributed in 4 clusters. It should be noted that other 4-cluster distributions of FM patients exist [12,35,36,37,38,39,40,41], but used other measures for cluster derivation (e.g., clinical measures related to FM symptomatology) reflecting 4 clusters of differing levels of global severity.

Overall, although men were underrepresented, our pooled study sample was representative of the population of patients with FM, who are primarily female, middle-aged, Caucasian, and have at least some college education. This pooled US sample used for the present study was optimally classified in 4 subgroups according to the HCA and confirmed by the LPA, as every FIQR subscale and the total score of the questionnaire presented significant differences between clusters with a large effect size (i.e., ηp^2^ > 0.14), always following the expected pattern: 1 < 2 < 3 < 4. Sociodemographic characteristics presented some statistically significant differences between subgroups, such as age, gender, or marital status, which should not be considered clinically significant in any case and most likely attributable to the large sample size, as for instance, the proportion of women in the 4 clusters ranged between 95.5% (cluster 1) and 97.4% (cluster 3), and ages ranged between 51.48 (clusters 3 and 4) and 53.24 years old (cluster 1).

The education level did show a between-subgroup difference that should be noted: According to our data, the level of education is inversely proportional to the FM severity. While 49.4% of FM patients in subgroup 4 (i.e., “severe”) had completed primary education, only 8.8% had post-graduate education; this proportion was notably higher in the other subgroups (1 = 26%, 2 = 21.2%, 3 = 14.8%). This result calls for a discussion on the sociodemographic profile of the FM patient; while it has been widely studied that FM affects mainly middle-aged women, not so much attention has been directed to the economic and educational profile of this population. Some previous studies have reported strong negative correlations between low levels of education and FM severity [42,43,44]. In addition, FM has been found to be more prevalent in people with low socioeconomic status in countries like Brazil [45], Turkey [46], or Canada [47], and low income was reported as a variable that increases the risk of suffering FM in a cohort study conducted in Finland [48]. There might be different coherent explanations for the effect of education on FM prevalence and its severity, yet it seems reasonable to include among them the type of jobs that low-educated people usually have access to (blue-collar workers), which are often physically demanding; this, along with other variables (e.g., genetic vulnerability, stressors, and other psychological comorbidities such as anxiety or depression), may trigger the dysregulation of the central nervous system, responsible for the symptomatology of FM (e.g., hyperalgesia and allodynia, among others) [2]. Additionally, those jobs are typically associated with lower income, which is an objective obstacle for accessing private health insurances and useful health-resources such as yoga classes or psychotherapies that are usually out-of-pocket costs in most countries [49].

Concerning external validity, the subgroups presented statistically significant differences in pain, anxiety symptoms, perceived cognitive impairment, fatigue, and sleep problems. Most of these differences presented a large effect size (ηp^2^ > 0.14), except for anxiety and perceived cognitive impairment, which presented medium effect sizes. The significant differences always followed the expected pattern (i.e., 1 < 2 < 3 < 4), although in some cases there were no significant differences between some of the subgroups. Surprisingly, the only variable that did not show any difference between subgroups was depressive symptoms, as measured by the HADS-D, whose scores were relatively low in all cases (i.e., ranging 7–8 out of 21), although participants in cluster 4 presented a notably higher standard deviation than the rest. However, when analyzing the FIQR item related to depressive symptomatology (item 16), significant differences were appreciated between clusters in the expected direction (*p* < 0.001). This result is consistent with previous findings, as depressive symptomatology has been reported to differ significantly among FM subgroups in various studies, including those using the FIQR as cluster derivation measure [11,13,35,37,39,50,51], for what it could be concluded that the HADS-D did not measure with precision the depressive symptomatology of this subsample. In this regard, some studies have indicated that the HADS does not differentiate well between depressive and anxious symptoms, suggesting that subscale scores are not warranted and that only the total score should be computed and reported [52]. Despite that counterintuitive result, and considering that the rest of the secondary measures did present the expected significant differences between clusters, a free online cluster calculator was created following the path drawn by Pérez-Aranda et al. [11], to share the results of the present study so future researchers can implement them to study how the clusters respond to different treatment alternatives and further extend studies into the characterization of FM. This calculator can be found as Appendix A.

The limitations of this study include, as stated previously, using only one measure for cluster derivation (i.e., the FIQR), which implies some shortcomings compared to other methods that include more measures. However, a comprehensive approach would be unaffordable for real clinical practice, in the case clusters could be used for personalized medicine. One of the major limitations in the present work was that diagnosis of FM was self-reported and not provided by medical records or confirmed by a healthcare professional. Another shortcoming in this work is that we do not know whether participants were under treatment at the time of the study. Another notable limitation of the present study is the relatively low proportion of participants that completed the measures for external validation, approximately a 4% of the pooled sample, which in the case of HADS-D was clearly not representative of the FM population. Moreover, it should be considered that the two samples conforming this study include a very low proportion of men with FM; this is a common shortcoming of cluster studies conducted to date, as FM was believed to affect mostly women. With the latest update of the diagnostic criteria from the American College of Rheumatology, estimates indicate men should represent almost one third of FM patients [53]. Furthermore, it is also known that fewer men respond to online recruitment in general [54].

## 5. Conclusions

The present study supports validation of a 4-cluster distribution based on FIQR scores for a large US sample of FM patients. The main challenge for future studies is now to test how the different subgroups respond to the already validated interventions in the interest of personalized medicine.

## Figures and Tables

**Figure 1 ijerph-18-00247-f001:**
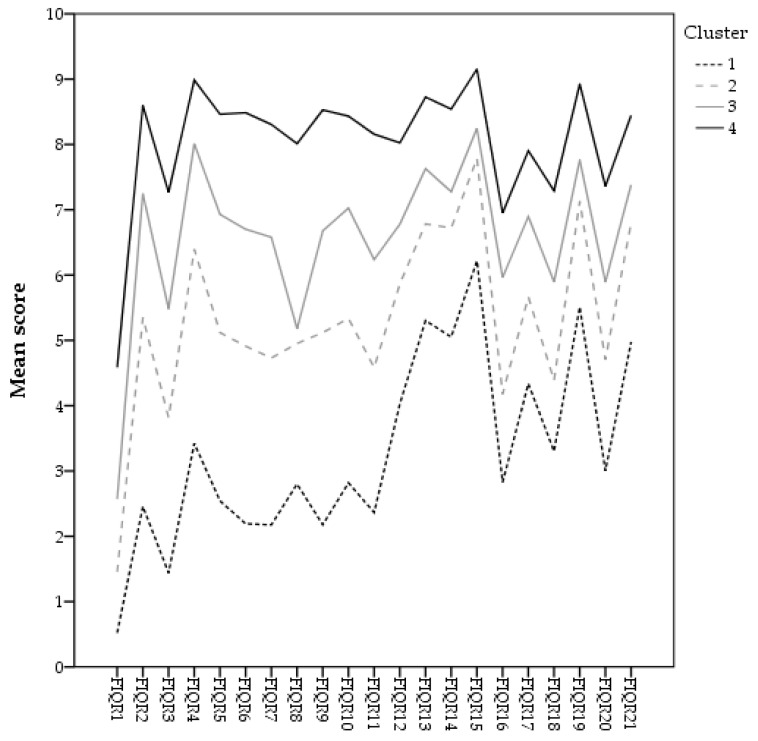
Hierarchical clusters distribution (calculator-derived).

**Table 1 ijerph-18-00247-t001:** Sociodemographic characteristics per Fibromyalgia Impact Questionnaire-Revised (FIQR) calculator-derived cluster.

Sociodemographic Characteristics	Cluster 1	Cluster 2	Cluster 3	Cluster 4	*p* Value
**Gender, N of females (%)**	1038 (95.2%)	1466 (96.6%)	1447 (96.7%)	2118 (97.3%)	0.009
**Age, M (SD)**	53.30 (11.78)	52.77 (11.06)	51.81 (10.67)	51.34 (9.80)	<0.001
**Marital status, N of married (%)**	623 (72.4%)	831 (72.1%)	750 (68.9%)	1049 (65.7%)	0.001
**Education, N (%)**					<0.001
Less than high school	1 (0.1%)	1 (0.1%)	3 (0.3%)	5 (0.3%)	
High school graduate	75 (8.7%)	96 (8.3%)	115 (10.6%)	252 (15.2%)	
College/trade school	247 (28.7%)	402 (34.9%)	482 (44.3%)	789 (49.4%)	
Bachelor’s degree	313 (36.4%)	409 (35.5%)	327 (30.1%)	420 (26.3%)	
Postgraduate	224 (26%)	244 (21.2%)	161 (14.8%)	140 (8.8%)	
**Years of Fibromyalgia (FM) diagnosis, M (SD)**	12.59 (7.77)	12.15 (7.54)	11.70 (7.30)	11.58 (7.63)	0.007

**Table 2 ijerph-18-00247-t002:** Clinical outcomes per FIQR calculator-derived cluster.

Clinical Outcomes	Cluster 1	Cluster 2	Cluster 3	Cluster 4	ηp^2^	Pairwise
**FIQR**						
Physical dysf.	6.32 (3.50)	13.63 (3.21)	18.57 (2.58)	23.92 (2.85)	**0.82**	1 < 2 < 3 < 4
Overall impact	5.03 (3.22)	9.72 (3.58)	13.23 (3.21)	16.76 (2.20)	**0.65**	1 < 2 < 3 < 4
Symptom sev.	22.01 (6.53)	29.89 (5.02)	34.54 (4.36)	41.01 (4.20)	**0.66**	1 < 2 < 3 < 4
Total score	33.36 (9.47)	53.25 (6.02)	66.34 (4.95)	81.69 (7.14)	**0.86**	1 < 2 < 3 < 4
**BPI**						
Severity	3.32 (1.53)	4.80 (1.31)	5.80 (1.11)	6.77 (1.15)	**0.50**	1 < 2 < 3 < 4
Interference	3.26 (1.63)	5.74 (1.35)	6.67 (1.46)	8.21 (1.05)	**0.63**	1 < 2 < 3 < 4
**HADS**						
Anxiety	9.97 (3.51)	11.92 (4.55)	12.97 (4.39)	13.14 (3.32)	**0.07**	1 < 3, 4
Depression	7.29 (2.62)	7.38 (2.36)	7.38 (2.99)	7.23 (4.02)	**0.00**	-
**MASQ**	119.27 (5.97)	119.89 (5.14)	121.44 (6.81)	123.47 (7.64)	**0.06**	1, 2 < 4
**MFI-20**						
General fatigue	16.41 (2.27)	17.67 (2.23)	18.72 (1.73)	19.34 (1.11)	**0.23**	1 < 2 < 3, 4
Physical fatigue	13.39 (4.48)	15.91 (2.90)	17.70 (2.37)	18.51 (1.80)	**0.28**	1 < 2 < 3, 4
Activity	12.34 (4.36)	13.89 (4.26)	15.71 (3.33)	17.17 (2.84)	**0.18**	1, 2 < 3 < 4
Motivation	12.03 (4.67)	14.32 (3.60)	15.77 (3.76)	17.01 (3.20)	**0.16**	1, 2 < 3, 4
Mental fatigue	10.94 (2.70)	14.03 (3.05)	14.97 (3.51)	16.47 (2.80)	**0.24**	1 < 2, 3 < 4
**MOS-Sleep**	19.43 (6.18)	23.87 (4.34)	25.09 (4.51)	26.90 (4.57)	**0.21**	1 < 2, 3
						1, 2 < 4

Note: In bold, statistically significant results (*p* < 0.05). Score ranges (min-max) appear between brackets in the first column. FIQR: Fibromyalgia Impact Questionnaire-Revised; BPI: Brief Pain Inventory; HADS: Hospital Anxiety and Depression Scale; MASQ: Multiple Ability Self-Report Questionnaire; MFI-20: Multidimensional Fatigue Inventory; MOS-Sleep: Medical Outcomes Study sleep scale.

## Data Availability

Data available on request due to privacy restrictions.

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
