# Peer review of "Subgrouping a Large U.S. Sample of Patients with Fibromyalgia Using the Fibromyalgia Impact Questionnaire-Revised"

_ijerph, 2020, doi:10.3390/ijerph18010247_

Round 1

Reviewer 1 Report

This an important study to improve care to FM patients. The aim of study is to proceed with the cross-validation of the four-cluster classification based on FIQR 85 scores in a large US sample (N= 6,280) of individual with FM. The study benefits from a large sample of patients with FM, which allows different analyses. This represents its most important strength.

A number of points deserve reflection and consideration.

The authors should address the possible bias due to some patients  were recruited based on a diagnosis by self-report.

Strategy taking regarding missing data in the different items of FIQR

Possible limitations of the use of the tool developed in clinical practice (ex. The authors don’t know the treatments that participants are performing)

Is the sample included representative of US population of patients with FM?

Author Response

Dear Ms or Mr,

Thank you very much for your comments and suggestions.

Find bellow a point-by-point response to your queries:

1. We agree that the possible bias due to self-reported diagnosis should be acknowledged; it has been added in the “Limitations” paragraph, lines 308-310:

One of the major limitations in the present work was that diagnosis of FM was self-reported and not provided by medical records or confirmed by a healthcare professional.

2. Missing data regarding Study 1 was detailed in lines 97-98. However, it is true that we did not make explicit that no missing data were found in Study 2. This information is now added in lines 105-106:

No missing data were found in our main study measure (FIQR).

3. We assume that part of the participants of the study were under treatment, and we consider that including all kind of FM patients–no matter what treatment they were or not receiving–was the best way of presenting a realistic picture of this population.

However, we have acknowledged this limitation as follows (310-311):

Another shortcoming in this work is that we do not know whether participants were under treatment or not at the time of the study.

4. Regarding the representativeness of our sample, as indicated in line 93, the participants of Study 1 were members of the National Fibromyalgia Association (i.e. included people from different states), and participants from Study 2 (lines 106-107) “represented all 50 states in the US”. This should be a solid indicator of representativeness of our sample, next to other sociodemographic data such as gender, age, ethnicity, and education level. This has been stated in lines 252-254:

Overall, although men were underrepresented, our pooled study sample was representative of the population of patients with FM who are primarily female, middle-aged, Caucasian and have at least some college education.

Although it comprises mostly women (around 97%), and therefore our findings could not be transferable to men with FM. This has been added as a limitation in lines 314-319:

Moreover, it should be considered that the two samples conforming this study include a very low proportion of men with FM; this is a common shortcoming of cluster studies conducted to date, as FM was believed to affect mostly women. With the latest update of the diagnostic criteria from the American College of Rheumatology, estimates indicate men should represent almost one third of FM patients [53]. Furthermore, it is also known that fewer men respond to online recruitment in general [54].

Reviewer 2 Report

I would like to thank the journal for the opportunity to review this manuscript. The manuscript is of great quality in relation to its conceptual framework, its methods and the discussion of the results, and I therefore recommend its publication in the International Journal of Environmental Research and Public Health.

I would like to highlight that its strong points are in relation to the large sample size (n=6625) and its clinical applications. As the authors point out, the heterogeneity of the patients with fibromyalgia is a widely known fact and there is, therefore, a great need to have tools that allow us to establish subgroups of patients what will allow us to classify them so that clinicians can more precisely prescribe adequate therapies for each patient. The manuscript also has as an advantage that the tool used for the creation of the subgroups is the FIQR, an instrument specifically designed for fibromyalgia patients and that is widely used in clinical settings as a measure of impact of the disease. Furthermore it’s simple to administer, and therefore the use of this instrument to create subgroups is another advantage to the current manuscript. Also, as the authors point out, with the use of this instrument (FIQR) the classification found in the manuscript in a sample of North Americans coincides with the classification found by the same research group, in a Spanish sample. In this sense, another advantage of this study resides in its transcultural nature. We would also like to thank the authors for having provided additional material, an excel application that allows for the calculation of subgroups that the patients belong to (within the four being considered).

The comments that are presented below (minor revision), are some of the suggestions that, from my point of view, could contribute to the improvement of the study, although it is already of very high quality.

  • The manuscript does not specify what diagnostic criteria were used to assess the patients with fibromyalgia in the sample. Although previous literature reveals a great concordance between both criteria for FM diagnosis (1990 and 2010 ACR; Carrillo-de-la-Peña et al., 2015), it is also true that the 2010 criteria are a step forward from the initial diagnostic criteria that only take pain as being the main symptom. While widespread musculoskeletal pain throughout the body and fatigability are the main symptoms of FM, a variety of extra-skeletal symptoms such as fatigue, sleep problems, asthenia, cognitive alterations, gastrointestinal problems, anxiety, depression, migraine or paresthesia, among others, are also included among the current diagnostic criteria (Wolfe et al., 2010). The relevance of these symptoms, in addition to pain, has led to a change of the original diagnostic criteria (ACR 1990 criteria), currently, besides pain, other symptoms need to be taken into account (ACR 2010, 2016 criteria) as part of the diagnostic criteria (Ahmed, Aggarwal, & Lawrence, 2019). This could affect the variability of the symptoms of the patients, so I suggest the authors reflect on this at some point in the manuscript.

  • Within the first sample used for the current study (n= 1,639) it is pointed out that “This study included clinical self-report measures that were used for the assessment of external validity of the subgroups in the present study (see measures used for external validation)”. In the instruments section, when the authors describe each of the measures and instruments used for the above mentioned external validation, different samples are specified for each instrument, ranging between n=252 for The Multiple Ability Self-Report Questionnaire; MASQ and n=285 for The Hospital Anxiety and Depression Scale; HADS. Nevertheless, the characteristics of these subsamples are unknown, how they were selected or if the subsample was the same for all measures or not, among other questions. Given the relevance of this subsample in the methods and results of the current study it would be of interest, additionally, to know the variability of this sample regarding the sociodemographic and clinical variables taken into account in relation to the total sample, as well as in relation to their scores on the FIQR and its dimensions.

  • Furthermore, in the same line as the above comment, it would be of interest to know the distribution of the subsample used for the external validation for each of the four clusters, so as to be able to contextualize the results.

  • In the discussion, as the authors point out, an interesting result is that “the only variable that did not show any difference between subgroups was depressive symptoms, whose scores were relatively low in all cases”. It could be that the contextualization of the subsample for ‘external validation’ that has been discussed above could offer some additional explanation in this sense (points 2 and 3). The large SD (SD= 4.02) in depressive symptoms of the cluster 4 in comparison to the rest of the SD for the rest of the groups (around 2.5) is also noteworthy.

  • Finally, given the important repercussions of the current manuscript and its undoubtedly important clinical repercussions, it would be appreciated if there were a small section in the discussion regarding the most adequate therapeutic indications for each particular type (cluster) of patients.

Author Response

Dear Ms or Mr,

Thank you very much for your comments and suggestions. 

Find bellow a point-by-point response to your queries:

1. Diagnosis was self-reported and not confirmed by a healthcare professional. This has been added as a limitation (lines 308-310):

One of the major limitations in the present work was that diagnosis of FM was self-reported and not provided by medical records or confirmed by a healthcare professional.

Nonetheless, this limitation does not compromise across-study comparisons. In fact, we have replicated the four FIQR clusters recently found by Pérez-Aranda et al (2019) in Spanish individuals with FM.

On the other hand, it has been stated that the 2010 ACR criteria could imply some significant changes in the composition of the samples, particularly in the gender variable. This information can be found in lines 314-319:

Moreover, it should be considered that the two samples conforming this study include a very low proportion of men with FM; this is a common shortcoming of cluster studies conducted to date, as FM was believed to affect mostly women. With the latest update of the diagnostic criteria from the American College of Rheumatology, estimates indicate men should represent almost one third of FM patients [53]. Furthermore, it is also known that fewer men respond to online recruitment in general [54].

2. As you indicate, each study measure used for cross-validation was answered by different subsamples of around 250-280 participants, which were generated at random and therefore should be representative of the entire sample. That is the reason why the characteristics of each subsample have not been described.

We have tried to clarify this issue in lines 100-101:

[…]; random subsamples of around 250-280 participants were generated to answer each self-reported measure.

3. Although analyzing the distribution of every subsample used for external validation could be undoubtedly interesting, this would imply describing up to 5 subsamples (one for each questionnaire: BPI, HADS, MASQ, MFI and MOS), all of them consisting of less than 300 participants, which would probably imply a very low representation of some of the clusters. Moreover, we believe that this information could deviate the attention from the main results, as the secondary outcomes were only added to find significant differences between the clusters, but with an exploratory intention.

4. In the case of the HADS-D subsample, we agree that adding information regarding its characterization could be very useful to clarify our results, but as counterintuitive as it may seem, no significant differences between this subsample and the global sample have been appreciated: the cluster distribution was relatively similar (i.e. C1= 7%, C2= 23%, C3= 30%, C4=39%) and there were no differences in sociodemographic data.

However, we have analyzed the FIQR item regarding depressive symptomatology (item 16), and in this case significant differences between clusters are observed (both with the total sample and with the subsample of 278 patients who had completed the HADS-D). This could be indicating that, despite its good psychometric properties, HADS-D was not assessing depressive symptomatology with precision in our subsample. It should be noted that due to the existence of a solid global factor of general distress, the HADS and other self-report measures such as the DASS-21, do not differentiate well between depressive and anxious symptoms, suggesting that subscale scores are not warranted and that only the total score should be computed and reported (Luciano et al., 2014).

These considerations have been added in lines 288-299:

Surprisingly, the only variable that did not show any difference between subgroups was depressive symptoms, as measured by the HADS-D, whose scores were relatively low in all cases (i.e. ranging 7-8 out of 21), although participants in cluster 4 presented a notably higher standard deviation than the rest. However, when analyzing the FIQR item related to depressive symptomatology (item 16), significant differences were appreciated between clusters in the expected direction (p < .001). This result is consistent with previous findings, as depressive symptomatology has been reported to differ significantly among FM subgroups in various studies, including those using the FIQR as cluster derivation measure [11,13,35,37,39,50,51], for what it could be concluded that the HADS-D did not measure with precision the depressive symptomatology of this subsample. In this regard, some studies have indicated that the HADS does not differentiate well between depressive and anxious symptoms, suggesting that subscale scores are not warranted and that only the total score should be computed and reported [52].

5. Your comment regarding which could be the most adequate indication for each cluster is very insightful and raises the one-million-dollar question: What treatments work and for whom? This remains as a major challenge when providing clinical care for FM.

From our point of view an optimal solution would be to establish some scalability of treatments depending on the severity of the patient. In other words, some ingredients of treatments (e.g. education and some counselling about aerobic exercise) should be universal, being sufficient for patients from the less impaired clusters (class 1 and 2), whereas other non-pharmacological ingredients (e.g. based on acceptance and mindfulness-based therapies) should be provided to those situated in the moderate-severe cluster (class 3). Finally, those patients situated in the most severe cluster (class 4) would need intensive forms of psychotherapy, physical therapy plus medication.

Nonetheless, there are some steps that need to be conducted first and that have not been produced yet, such as performing randomized controlled trials comparing the efficacy of different treatments for different FM clusters. Without this information, adventuring which treatment could be more indicated for each cluster would be completely hypothetical.

Reviewer 3 Report

Dear authors,

Congratulations on the manuscript. It provides an interesting approach to the analysis and the of the FM using the revised FIQ.

The approach of using the same Questionnaire in a new context is always a challenge due to the contextual and cultural differences.

There are a series of comments regarding the manuscript:

  • There is a concern on the low response rate and an extended explanation would be appreciated.
  • Lines 229-230. The statement “the present study fills this gap by replicating the proposed …” is contradicted by lines 301-302 where it is stated the low proportion response.
  • Lines 235-236. The sentence “unlike previous studies…” requires a reference to those previous studies.
  • Lines 244-246. The sentence implies things that are not proven “It is possible that the different results are due to the size of the samples i.e. hypothetically…”. Please review, reference it or discard it.
  • Lines 286-290. Would you so kind as to provide an explanation of this counterintuitive result?.
  • Lines 301-302. The low proportion of participants is concerning. As this is the discussion area, It would be appreciated to compare it with examples and proportions obtained in other studies.
  • References: 26 out of the 51 references are over 10 years old. Some of them are already in the year 2000, but 10 of them are even older.

Best regards.

Author Response

Dear Ms or Mr,

Thank you very much for your comments and suggestions.

Find bellow a point-by-point response to your queries:

1. We believe that the low response rate issue that you point out is more of a misunderstanding than a real problem of our study; while all the subjects (6,280) completed the FIQR, the secondary outcomes (HADS, BPI, MOS, MASQ and MFI) were completed by random subsamples pointedly generated from Study 1. Following the design of the original study, different subsamples of around 250-280 FM patients responded each of those questionnaires.

We have tried to clarify this point in lines 100-101:

This study included clinical self-report measures that were used for the assessment of external validity of the subgroups in the present study (see measures used for external validation); random subsamples of around 250-280 participants were generated to answer each self-reported measure.

2. As abovementioned, we believe that the low proportion of response of secondary outcomes should not interfere with the main result of our study, which is the validation of the 4-cluster solution for FM patients in a large US sample. The secondary measures, such as HADS, BPI, etc., were used as an additional and exploratory aim of external validation.

3. Regarding the "Unlike previous studies..." incomplete sentence, we agree on that point; an example has been added. See line 238.

4. The statement regarding the relation between sample sizes and number of clustesr, while hypothetical, has empirical support as the studies conducted with larger samples tend to yield a higher number of clusters. Some examples are Wilson et al. (2009) (N = 2,583 à 4 subgroups) and Rehm et al. (2010) (N= 3,055 à 5 subgroups).

            Rhem, S.E., Koroschetz, J., Gockel, U., Brosz, M., Freynhagen, R., Tolle, T.R., Baron, R. A cross-sectional survey of 3035 patients with fibromyalgia: subgroups of patients with typical comorbidities and sensory symptom profiles. Rheumatology 2010; 49: 1146-1152.

            Wilson, H.D., Robinson, J.P., Turk, D.C. Toward the identification of symptom patterns in people with fibromyalgia. Arthritis Rheum 2009; 61: 527-534.

5. The HADS-D result is undoubtedly a strange result which we have tried to interpret from different angles; another reviewer suggested studying the characteristics of the subsample of patients who answered the HADS-D in order to detect some possible differences with respect to the total sample, but these were not found. We decided to analyze the FIQR item “depressive symptomatology” to see if this item presented similar results to the HADS-D, and it did not: significant differences, in the expected direction (i.e. cluster 4 had the highest score, and cluster 1 the lowest) were observed in this item. That leads us to conclude that, despite the strong psychometric properties of the questionnaire, the HADS-D did not work properly in this subsample.

This explanation has been added to the manuscript in lines 288-299:

Surprisingly, the only variable that did not show any difference between subgroups was depressive symptoms, as measured by the HADS-D, whose scores were relatively low in all cases (i.e. ranging 7-8 out of 21), although participants in cluster 4 presented a notably higher standard deviation than the rest. However, when analyzing the FIQR item related to depressive symptomatology (item 16), significant differences are appreciated between clusters in the expected direction (p < .001). This result is consistent with previous findings, as depressive symptomatology has been reported to differ significantly among FM subgroups in various studies, including those using the FIQR as cluster derivation measure [11,13,35,37,39,50,51], for what it could be concluded that the HADS-D did not measure with precision the depressive symptomatology of this subsample. . In this regard, some studies have indicated that the HADS does not differentiate well between depressive and anxious symptoms, suggesting that subscale scores are not warranted and that only the total score should be computed and reported [52].

6. As explained previously, that percentage of response was chosen by the researchers, who thought that subsamples of around 250-280 FM patients for each questionnaire would be methodologically sufficient for the analyses they wanted to perform in their study (Study 1). In our case, these analyses were secondary and exploratory, and that is the reason why no more attention has been granted to them.

7. Thank you for pointing the issue regarding the references. Some older studies were included in the text selectively because they are historically relevant, providing a good background in this research field. On the other hand, we also include the most recent references to studies conducted in the research line of subgrouping FM patients.
